# Chromoanagenesis from radiation-induced genome damage in *Populus*

**Weier Guo**, **Luca Comai**, **Isabelle M. Henry***

Genome Center and Dept. Plant Biology, University of California Davis, Davis, California, United States of America

* imhenry@ucdavis.edu

## Abstract

Chromoanagenesis is a genomic catastrophe that results in chromosomal shattering and reassembly. These extreme single chromosome events were first identified in cancer, and have since been observed in other systems, but have so far only been formally documented in plants in the context of haploid induction crosses. The frequency, origins, consequences, and evolutionary impact of such major chromosomal remodeling in other situations remain obscure. Here, we demonstrate the occurrence of chromoanagenesis in poplar (*Populus sp.*) trees produced from gamma-irradiated pollen. Specifically, in this population of siblings carrying indel mutations, two individuals exhibited highly frequent copy number variation (CNV) clustered on a single chromosome, one of the hallmarks of chromoanagenesis. Using short-read sequencing, we confirmed the presence of clustered segmental rearrangement. Independently, we identified and validated novel DNA junctions and confirmed that they were clustered and corresponded to these rearrangements. Our reconstruction of the novel sequences suggests that the chromosomal segments have reorganized randomly to produce a novel rearranged chromosome but that two different mechanisms might be at play. Our results indicate that gamma irradiation can trigger chromoanagenesis, suggesting that this may also occur when natural or induced mutagens cause DNA breaks. We further demonstrate that such events can be tolerated in poplar, and even replicated clonally, providing an attractive system for more in-depth investigations of their consequences.

## Author summary

Plant breeders often use radiation treatment to produce variation, with the goal of identifying new varieties with superior traits. We studied a population of poplar trees produced by gamma irradiation of pollen, and asked what kind of DNA changes were associated with this variation. We found many changes, most often in the form of added (insertions) or removed (deletions) pieces of DNA. We also found two lines with much more drastic changes. In those lines, we observed massive reorganization. We characterized these two lines in detail and found that catastrophic pulverization and random reassembly only occurred on a single chromosome. Looking closely at how the pieces were put back together suggest that the rearrangements in these two lines may have resulted from two

**Data Availability Statement:** The sequences reported in this paper have been deposited in the National Center for Biotechnology Information BioProject database (BioProject ID: PRJNA723573).

**Funding:** This research was supported by the US Department of Energy Office of Science, Office of Biological and Environmental Research, Grant DE-SC0007183 to LC (https://www.energy.gov/science/ber/biological-and-environmental-research). The DNA Technologies and Expression Analysis Core at the UC Davis Genome Center is supported by National Institute of Health (NIH) Shared Instrumentation Grant 1S10OD010786-01 (https://www.nih.gov/grants-funding). The funders had no role in study design, data collection and analysis, decision to publish, or preparation of the manuscript.

**Competing interests:** The authors have declared that no competing interests exist.

slightly different mechanisms. This type of rearrangement is commonly observed in human cancer cells, but has rarely been observed in plants. We demonstrated here that they can be induced by gamma irradiation, indicating this type of event might be more widespread than we expected. Characterizing such genome restructuring instances helps to understand how genome instability can remodel chromosomes and affect genome function.

## Introduction

Genomic structural variation (SV) includes various types of chromosomal rearrangements, such as insertion, deletion (INDEL), copy number variation (CNV), inversion and translocation. Structural variation can produce evolutionary significant variation, because it can affect large regions of the genome, and influence multiple traits at once. In one extreme scenario, restructuring of the genome results in clustered CNV affecting a single or a few chromosomes, a syndrome called chromoanagenesis. Chromoanagenesis results from a single triggering event that leads to highly complex segmental rearrangements [1,2]. The extreme restructuring of a single chromosome (or rarely two or more) results from two distinct processes: (i) in chromothripsis dsDNA breaks and Non Homologous End Joining rearrange tens to hundreds segments, with oscillations between two copy number states (occasionally three) [3,4], and (ii) in chromoanasynthesis, replication forks stalled at DNA breaks switch templates, resulting in segmental duplication and triplication events combined with complex chromosomal rearrangement of the implicated and intervening segments [5]. Chromothripsis and chromoanasynthesis are associated with missegregation of chromosomes, followed by micronucleus formation around a single chromosome, leading to a single, catastrophic pulverization event [6]. A third type of restructuring classified under chromoanagenesis differs in mechanism and outcome: during chromoplexy, chromosomes are broken in pieces, shuffled together and reassembled, resulting in rearranged chromosomes. Chromoplexy always affects more than one chromosome [7]. Chromoplexy can occur sequentially and may be originally related to DNA breaks caused by transcription factor binding [8]. In plants, chromoplexy-like events have been observed in natural variants in camelina [9], and also as a consequence of plant transformation in arabidopsis, rice and maize [10].

Chromothripsis and chromoanasynthesis were originally identified in human cancerous cells [1]. To distinguish them from indels, precise criteria are applied [11,12]. In plants, there are multiple cases of extensive genomic rearrangements [10,13], but when applying the important criterion of highly frequent and clustered (at least 10) rearrangements within a single chromosome, only haploid induction crosses in *Arabidopsis thaliana* display catastrophic chromosomal reconstructing patterns [14]. In these haploid induction crosses, both chromothripsis and chromoanasynthesis were detected, and the early zygotic divisions are often also accompanied by the formation of micronuclei [15], another diagnostic feature of chromothripsis [1,12].

A critical step in the plant life cycle is pollen production and fertilization. Pollen is prone to natural mechanisms that break DNA [16,17] and it is also a classical target for chemical and radiation mutagenesis [18]. While traditional chromosomal rearrangements have been described, the range of variation resulting from these mechanisms, however, has not been determined. We decided to address this question in a poplar F1 population that we previously developed from an interspecific cross using gamma-irradiated pollen. This population was characterized genetically and harbors >650 unique large-scale insertions and deletions, ranging from a few hundred kbp to entire chromosomes. Cumulatively, these indels cover the genome multiple times [19]. To investigate whether gamma irradiation could have also

resulted in more severe genome reorganization events, we screened this population for signs of clustered copy number variation patterns. We identified two individuals with genomic patterns reminiscent of chromoanagenesis, which we characterized in detail. Our results indicate that DNA breaks induced by irradiation triggered single chromosome fragmentation and restructuring patterns consistent with chromoanagenesis. These results suggest that pollen DNA breaks, either natural or induced, can produce extreme structural variations that may provide evolutionary innovation and, in perennial plants such as poplar, where we were able to preserve the chromoanagenesis outcomes by vegetative propagation, provide an attractive system for long-term investigation of the outcome of chromoanagenesis.

## Results

### Gamma irradiation can result in chromoanagenesis in poplar

A poplar *P. deltoides* x *P. nigra* F1 hybrid population was developed previously [19], and characterized using low-coverage illumina genome sequencing. In this population, ~58% of the lines carried large-scale genomic insertions and deletions (indels) [20], induced by gamma-irradiation of pollen grains before fertilization. Each F1 line was characterized by a unique set of indels randomly distributed along the 19 chromosomes of the poplar genome [19,20]. Interestingly, two of these lines exhibited dosage variation consistent with chromoanagenesis. Specifically, they displayed multiple clustered CNVs on a single chromosome. To investigate the mechanisms that resulted in these extreme genomic rearrangements, we selected 9 lines for further analysis: the 2 lines exhibiting extreme rearrangements (Shattering Group, Fig 1C), 4 lines with limited number of indels (Lesion Group, Fig 1B), and 3 lines with no apparent dosage variation (No-lesion Group, Fig 1A). Genomic DNAs from these 9 lines were sent for higher coverage Illumina genomic sequencing (coverage 25–50), with the goal of characterizing dosage variation in detail, especially those lines with shattered chromosomes.

The dosage variation patterns obtained using the deep-sequencing reads were consistent with their corresponding low-coverage data. Also consistent with previous results [19], parental allele frequencies from our high-coverage data indicated that all indels in the genome of the 9 selected individuals originated from loss or gain of the paternal *P. nigra* copy (Fig 2), confirming that the irradiated *P. nigra* pollen caused dosage variation.

Both lines in the Shattering Group fit our definition of clustered changes (>10 events per chromosome arm) (Fig 2A and 2B and S2 Table). POP33_31 exhibited 21 CNVs on Chromosome 1, including 2 deletions and 19 insertions, of sizes ranging from 10kb to 5.7Mb (Fig 2A and S2 Table). Among the clustered CNVs, we observed multiple copy number states, ranging from 1 to 5 (Fig 2A and S2 Table), suggesting that some fragments had been lost, while others went through duplication, triplication or even quadruplication. The second individual in the Shattering Group, POP30_88, only exhibited single-copy dosage variation. Specifically, 11 CNVs were found in this line, including 3 deletions and 8 duplications, all localized on Chromosome 2. These fragments ranged in size, from 80kb to 10.7Mb (Fig 2 and S1 and S2 Table). Taken together, these results suggested that chromoanagenesis is a possible outcome of gamma-irradiation. The different copy number variation patterns observed in these two lines further suggest that these two rearranged genomes might have been shaped by different rearrangement mechanisms.

### Novel DNA junctions can be detected using high-coverage short-read sequencing

To further confirm the hypothesis that these two lines underwent chromoanagenesis, we aimed to characterize their genome structure in detail (Fig 3). Specifically, we sought to

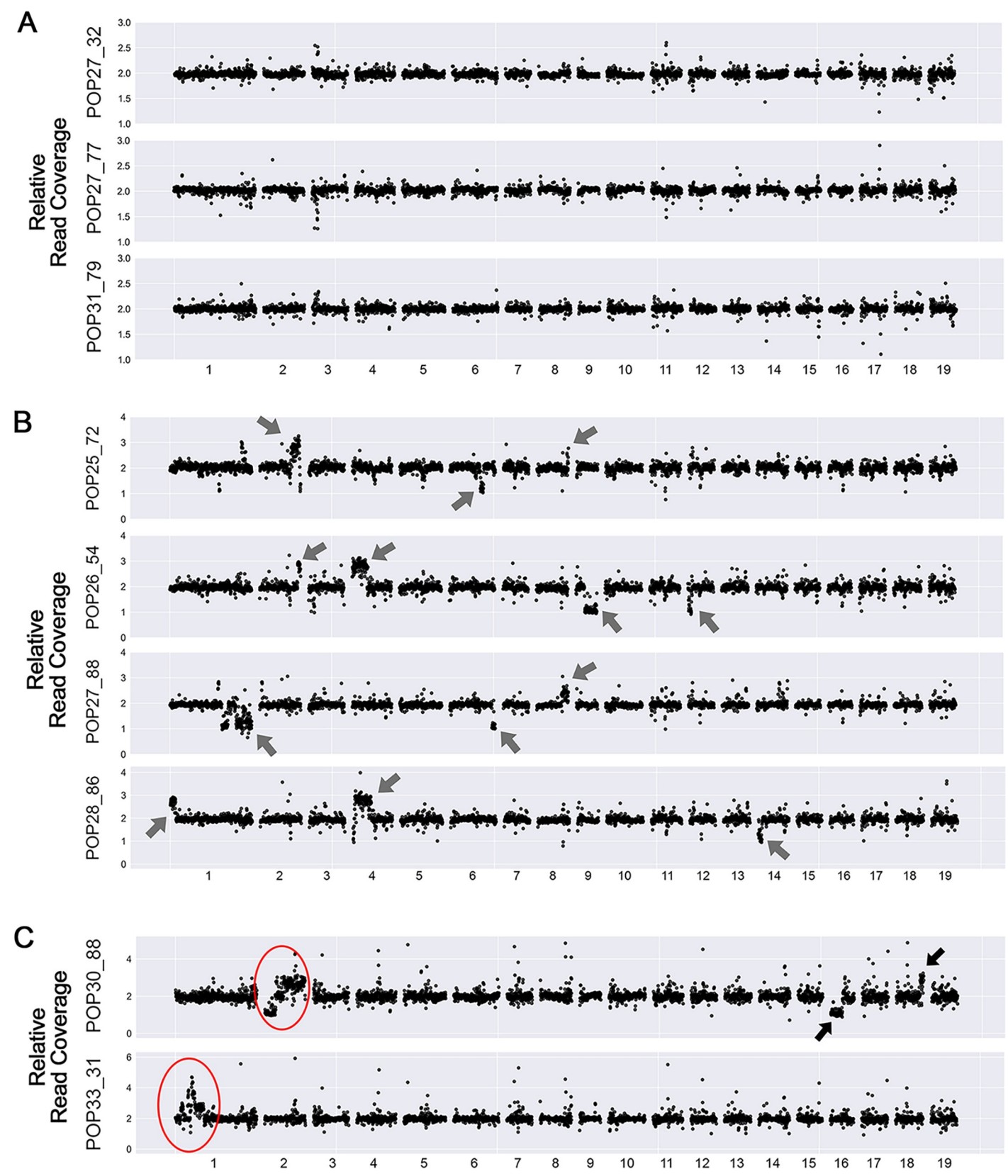

**Fig 1. Dosage variation detection.** Dosage variation was detected by displaying relative read coverage. Each data point represents the mean read coverage in non-overlapping 100kb bins, standardized to the mean read depth across all 9 lines. The expected value for a diploid line is a relative read coverage of 2. Values around 1 suggest deletions and values around 3 suggest insertions. (A) Dosage plots for 3 lines exhibiting no obvious instances of dosage variation. (B) Dosage plots for 4 lines containing a small number of indels. The arrows point to the randomly distributed indels identified. (C) Dosage plots for the 2 lines exhibiting shattering patterns. The red circles represent the regions displaying highly clustered copy number variation.

characterize the patterns of genome restructuring by searching for novel DNA junctions created with the observed rearrangements. To identify these novel junctions, we searched for sequencing reads with ends that mapped to two different positions within the genome, suggesting that these two sequences are now adjacent in the reconstructed genome. Because these rearrangements are expected to occur randomly, these junctions should be unique to each line. The boundaries of the indels described above provide prime candidates for the localization of novel junctions, but other locations in the genome are possible as well. Once potential junctions were identified, the exact position of the breakpoints were determined through *de novo* assembly of the corresponding sequencing reads.

Consistent with our expectations, multiple potential junctions were identified from both of the lines exhibiting shattered chromosomes, but overall fewer were identified for the other lines (Table 1). We next validated the presence of these junctions using PCR amplification followed by Sanger sequencing, and using sibling F1 lines as negative controls. For the two lines in the Shattering Group, 26/33 assembled potential junctions were validated by PCR. On the other hand, none of the potential junctions from the Lesion Group (0/22) and No-lesion Group (0/11) were validated. Junctions were determined as invalid if they could be amplified from the genome of other sibling lines as well, or if the Sanger sequencing results were not consistent with the expectation. In total, we identified 26 novel DNA junctions, all of which originated from the two shattered lines (S3 Table).

## Extreme genomic rearrangements are associated with intra-chromosomal junctions

By using the junction detection approach mentioned above, we observed multiple novel DNA junctions in the lines containing a shattered chromosome (Fig 4A). We next sought to characterize them further and attempted to reconstruct the rearranged sequences.

First, we documented the genomic localization of the validated junctions in each line. Junctions and dosage variation data were plotted together on Circos Plots (Fig 5). For both of the lines exhibiting shattering, all of the junctions were located on a single chromosome, whether they corresponded to a shift in dosage variation or not (Fig 5A and 5B). In POP33_31, only 2 breakpoints (each junction consisted of two breakpoints) occurred on regions with no detected dosage variation, while the other 22 overlapped with CNV boundaries (Fig 5C and S3 Table). However, in POP30_88, only 12/28 breakpoints corresponded to CNV regions (Fig 5D and S3 Table). This suggests that the mechanisms underlying the rearrangements in these two lines might differ. Based on the orientation of two junction ends, we observed that 17% and 36% of the junctions involved an inverted fragment in POP33_31 and POP30_88, respectively (S3 Table). Finally, we observed three different types of junctions based on the sequence structure of each junction: microhomology, perfect joining, and insertion (Fig 4B, 4C and 4D). Both shattered lines exhibited all three junction types (Fig 4A).

With the exact genomic position and orientation of two breakpoints in each junction, we were able to partially reconstruct the structure of the rearranged sequences in the two shattered lines. Specifically, we were able to construct 9 and 12 rearranged chromosomal pieces for POP33_31 (Chromosome 1) and POP30_88 (Chromosome 2), respectively (Figs 6 and S1). In

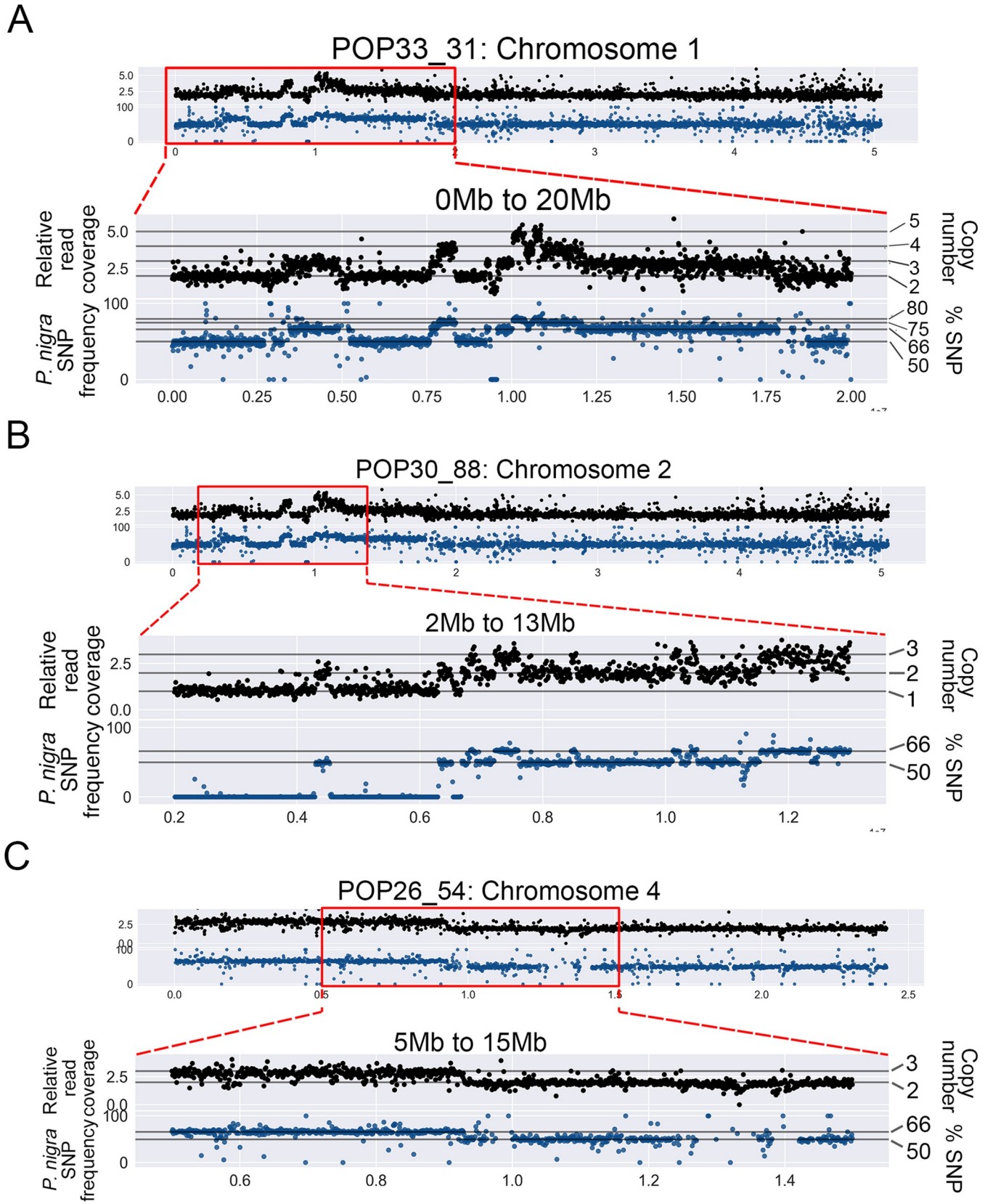

**Fig 2. Association of dosage variation patterns with SNP frequency.** To obtain a detailed view of the shattered regions, the genome was divided into narrower bins (10kb bins). Additionally, to confirm the origin of indels, *P. nigra* (male) SNPs frequencies were calculated for 10kb bins. Black scatterplots: each black dot represents the relative read coverage for a 10kb bin. Blue scatterplots: each blue dot represents the average *P. nigra* SNP frequency for a 10kb bin. Horizontal lines indicate the expected SNP frequency for different copy number states, as indicated on the right. (A) Chromosome 1 of POP33_3 displayed extremely clustered dosage variation within the first 20Mb, and all variation patterns were associated with *P. nigra* SNP frequency shifts. (B) Chromosome 2 of POP30_88 displayed extremely clustered dosage variation in the region between 3Mb and 13Mb, and all CNVs were associated with expected *P. nigra* SNP frequency shifts. (C) One of the large-scale lesions on the POP26_54 genome is shown, providing a comparison between larger randomly distributed indels and the observed shattering patterns in the other two lines.

both cases, our results suggest that the restructured region underwent extreme fragmentation, with chromosomal fragments joined together in a seemingly random order, some fragments lost altogether, and others copied multiple times (Figs 6 and S1).

## The novel DNA junctions are enriched in gene-rich regions

To investigate the DNA context around the novel junctions identified in the shattered lines, we asked whether the junctions occurred more often in genic or repeated regions of the genome. Every validated novel junction contained two breakpoints. For each breakpoint, we calculated the enrichment ratio (see Material and Methods) of genomic features. We used two different window sizes, 10kb and 100kb, for investigating gene contents and repeated elements. For both of the shattered lines, breakpoints occurred significantly more often in gene rich regions and significantly less often near repeated elements (Fig 7 and S6 Table). These results were consistent with previous studies of aneuploid *Arabidopsis thaliana* individuals carrying shattered chromosomes, which also indicated that breakpoints were more likely to occur in gene-rich regions [14]. Additionally, 26 breakpoints (50%) occurred within a gene coding sequence (11/24 for POP33_31; 15/28 for POP30_88, S4 Table), and 14 of these involved gene to gene fusion (S5 Table). Breakpoints were found on different genic elements, including coding region, introns and untranslated regions. Genes of various functions were affected by these breakpoints (meiosis-specific proteins, dynamin, etc, see S4 Table). These results indicate that novel DNA junctions induced by irradiation had the potential to dramatically influence the function of multiple genes at once.

## Discussion

We identified and characterized two instances of highly clustered CNVs on a single chromosome in poplar F1 hybrids that resulted from interspecific crosses using gamma-irradiated pollen. To investigate the structure of these extreme genome rearrangements, we characterized the candidate chromosomes from two individuals, and identified localized shattering and rejoining of DNA in each. Specifically, we identified and characterized 12 and 14 novel DNA junctions in these two lines, which were clustered on a single chromosome, and always appeared in the shattered genomic region. These observations are consistent with the characteristics of chromoanagenesis, which is a catastrophic event creating large numbers of complex rearrangements on one or a few chromosomes [1]. They also suggest that gamma-irradiation of pollen can result in chromoanagenesis-like patterns in poplar. In our population, we observed shattered chromosomes in 2/592 individuals. The two poplar lines carrying the shattered chromosomes did not exhibit significant phenotypic differences compared to their siblings. One of the two was sufficiently robust to be selected amongst the F1 individuals that were clonally propagated and transferred to a field for a population-wide phenotyping experiment [20,21], and did not exhibit extreme phenotypic behaviors in the traits analyzed.

To date, extreme chromosomal rearrangement have only been reported in a few plant species, including in aneuploid *Arabidopsis thaliana* individuals originating from haploid

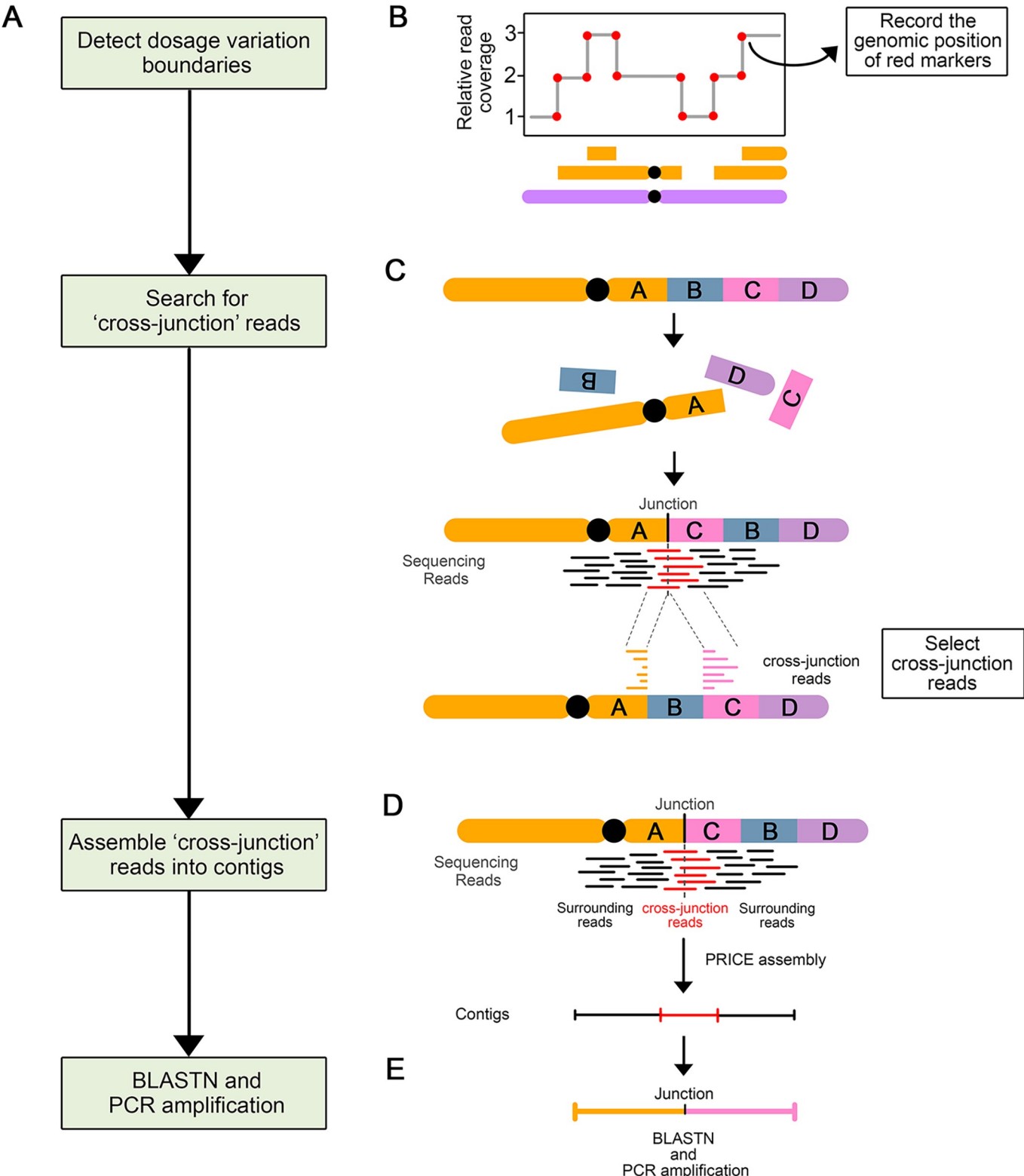

**Fig 3. Process of novel DNA junctions selection and validation.** (A) Flow chart illustrating the steps involved in novel DNA junction detection, selection, and validation. (B-E) Diagram illustrating the approach involved in each step. (B) A schematic dosage plot showing a genomic region containing many instances of dosage variations. The red dots highlight the boundaries of every indel and constituting potential breakpoint positions. (C) Schematic diagram illustrating the origin and mapping behavior of cross-junction reads. After chromosomal rearrangement, fragment A and C joined together and formed a novel DNA junction.

The sequencing reads (in red) that crossed this novel DNA junction are called cross-junction reads. These cross-junction reads map onto two different locations on the reference genome. (D) Assembly of the novel DNA junctions. Cross-junction reads are assembled into one contig using the PRICE assembler. (E) Each newly assembled scaffold is compared to the reference genome using BLASTN to: (i) find out the exact alignment positions of two breakpoints of the novel junction; (ii) confirm the uniqueness of contigs.

induction crosses [14], in maize and rice individuals that have undergone biolistic transformation [10], and in somatic variants of grape [9]. But, except for Tan's reports in *Arabidopsis*, which reported the observation of extreme DNA damage on a single chromosome, other reports described genomic restructuring involving multiple chromosomes and thus fitting chromoplexy [7]. Our study and Tan's study are the only two that showed evidence of clustered, single chromosomal rearrangement in plants, thus fitting the definition of chromothripsis and chromoanasynthesis [4,5].

The two shattered poplar lines both carry highly clustered breakpoints but differ in other ways, suggesting that the mechanisms underlying these events might be different. The line carrying a shattered Chromosome 1 (POP33_31), exhibits a wide variation in copy number states, ranging from 1 to 5, which indicates segmental duplication and triplication during the genomic remodeling. This is consistent with the replication-based complex rearrangements of chromoanasynthesis [5]. During chromoanasynthesis, the replication fork stops, and the polymerizing strand switches to a proximate template with micro-homologous sequences, and finally causes the formation of a complex chromosomal rearrangement involving multiple copy number states [22]. On the other hand, several features of the shattered chromosome of POP30_88 suggest that it is more likely to be the result of chromothripsis, the fragmentation and random reorganization of one or a few chromosomes [23]. First, the shattered chromosome of POP30_88 only exhibits three copy number states (1, 2 or 3), which is consistent with the limited copy number states observed in chromothripsis. Chromothripsis usually exhibits two copy number states: the lower one represents fragment deletion, and the higher one represents fragment retention [3]. Occasionally, it can also carry three copy number states. This can be caused by the partial duplication of the rearranged chromosome after experiencing chromothripsis [3]. The oscillation of three copy number states in POP33_88 Chromosome 2 suggests that it may have undergone chromothripsis, followed by a segmental duplication. Second, the novel DNA junctions observed in POP30_88 cover all four types of the orientations (H-T, T-H, H-H, T-T), and the rearranged fragments order appears random. This feature can also be potential evidence for chromothripsis, since the randomness of fragments

**Table 1. Summary of DNA junction validation frequencies.**

| Type | Lines | In silico Assembled Junction | PCR validated Junction | Validation Frequency | Assembled Junction in Group | Validated Junction in Group | Validation Frequency in Group |
|---|---|---|---|---|---|---|---|
| Shattering Group | POP33_31 | 16 | 12 | 0.75 | 33 | 26 | 0.788 |
| | POP30_88 | 17 | 14 | 0.824 | | | |
| Lesion Group | POP25_72 | 5 | 0 | 0 | 22 | 0 | 0 |
| | POP26_54 | 1 | 0 | 0 | | | |
| | POP27_88 | 14 | 0 | 0 | | | |
| | POP28_86 | 2 | 0 | 0 | | | |
| No-lesion Group | POP27_32 | 10 | 0 | 0 | 11 | 0 | 0 |
| | POP27_77 | 1 | 0 | 0 | | | |
| | POP31_79 | 0 | 0 | 0 | | | |

The validation frequencies represent the percentage of PCR-validated junctions out of the total number of in silico predicted junctions.

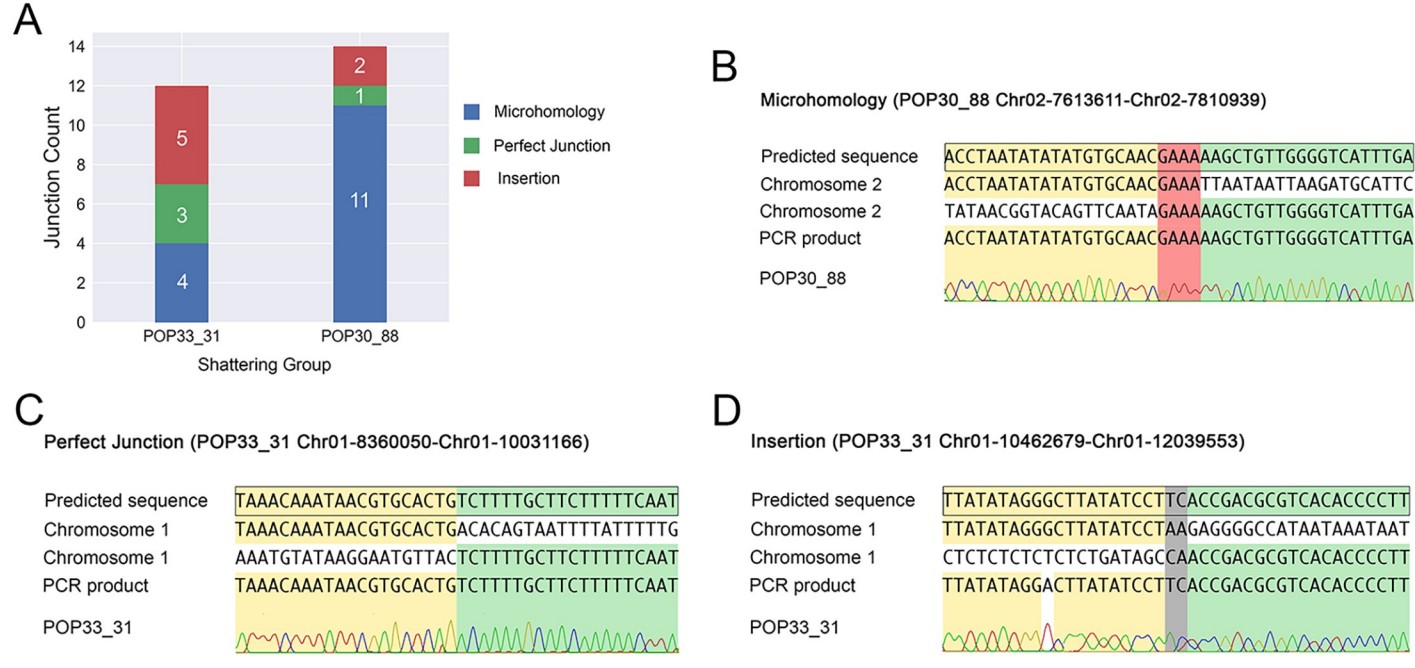

**Fig 4. Types of novel DNA junctions.** (A) Number and types of validated DNA junction identified in each line. Different colors represent the three junction types. (B) Microhomology: presence of 1–11 bp of overlap between the two reconstructed fragment ends. (C) Perfect junction: the two fragment ends are perfectly joined together, with neither overlapping bps nor inserted bps. (D) Insertion: 1–18 bp of novel nucleotide sequence is inserted between two fragment ends.

orientation and order is a representative property for this type of catastrophic event as well [4]. Altogether, our results suggest that the two chromosomal rearrangements observed might have originated from two different mechanisms: chromoanasynthesis for POP33_31 and chromothripsis for POP30_88.

Ionizing radiation has a long-standing role in plant mutation breeding [24]. The genomic consequences of ionizing mutations depend on tissue type [25], radiation dosage, and type of ionizing mutations, and can produce many different types of mutations [19,26–28], including the creation of variants exhibiting potentially favorable characteristics [20,21]. Ionizing radiation has also been proposed as a potential trigger of chromoanagenesis [3,29,30]. Finally, localized ionizing radiation targeting the nuclei of tumor cells was shown to induce chromoanagenesis-like patterns in those cells [31]. In this experiment, the authors used a microbeam system to precisely target the nuclei and induce double strand breaks in some chromosomes. Their study reported 14 *de novo* junctions involving four chromosomes, and proposed that targeted irradiation induced chromothripsis on a few chromosomes. Based on the features of the three types of chromoanagenesis events, Morishita's results suggest that their lines underwent chromoplexy, since the novel junctions are sparsely distributed on several chromosomes. Yet, it is also possible that, if the beam only targets a portion of the nuclei, only the chromosomes located in the affected area underwent rearrangement.

In contrast, our study reported highly clustered novel DNA junctions in a limited genomic region, while the initial irradiation treatment targeted whole desicated pollen grains [19]. Formation of extreme rearranged chromosomes by chromoanagenesis occurs over at least two mitotic divisions: during the first mitosis, a broken chromosome lags during anaphase and is incorporated into a micronucleus. During the following interphase, DNA replication of the micronucleus chromosome is delayed compared with the chromosomes in the major nucleus.

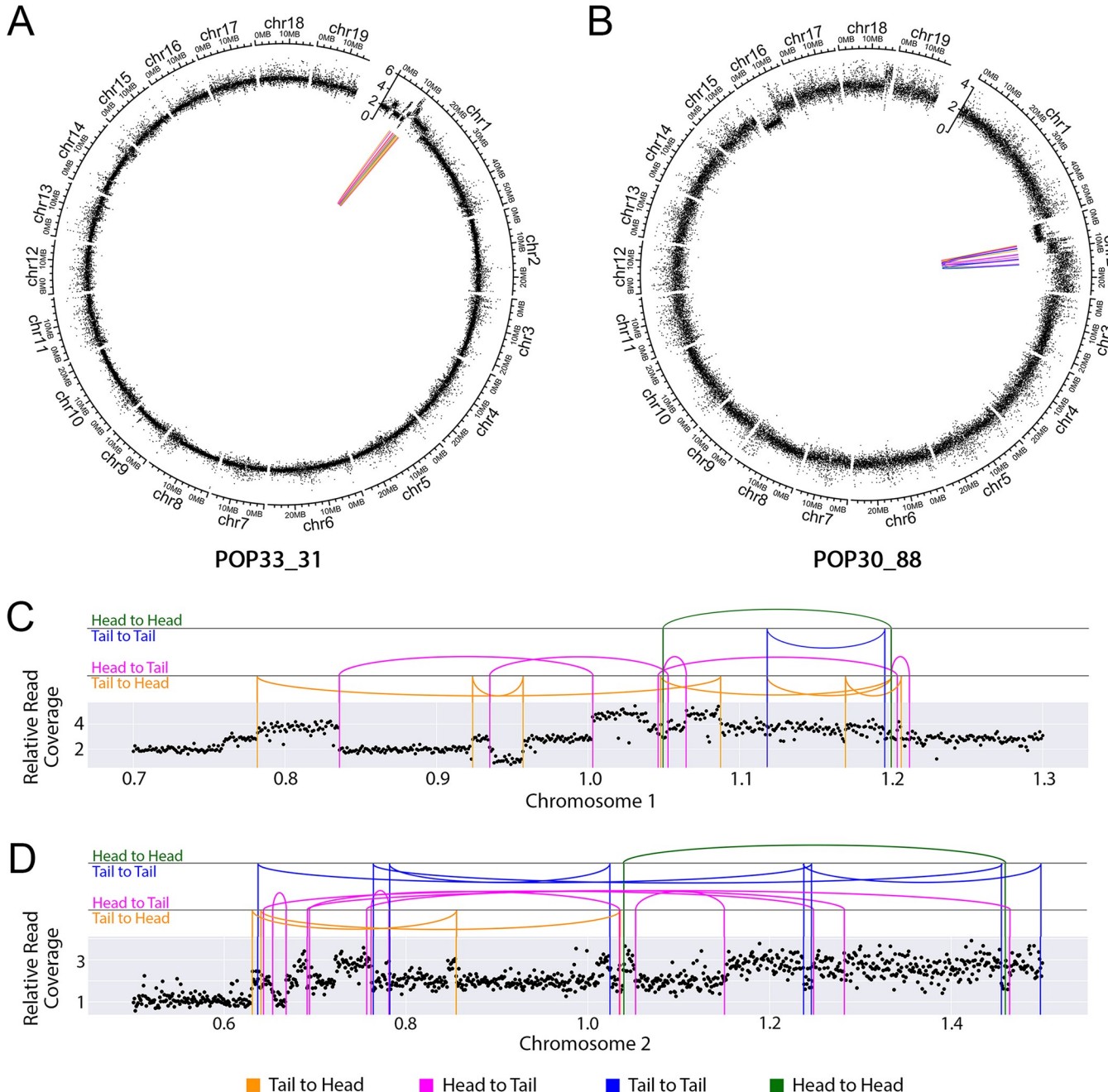

**Fig 5. Distribution of the genomic location of the validated DNA junctions.** (A-B) DNA junctions in the two shattered lines (POP33_31 (A) and POP30_88 (B)). The outermost layer displays each chromosome. The next layer displays relative reads coverage, averaged over 10kb non-overlapping bins. In the center, colored lines connect the original genomic locations of each pair of sequences found in novel DNA junctions. (C-D) Close-up view of DNA junctions distribution on the shattered regions of chromosome 1 in POP33_31 (C) and chromosome 2 in POP30_88 (D). The scatter plots show average relative read coverage per 10kb bins, and the colored vertical lines represent exact breakpoints. The arc connecting two vertical lines illustrate the novel junctions connecting vertical lines that represent the breakpoints. All panels: Magenta and orange lines represent sequences that connect in the same direction (Head to Tail in magenta and Tail to Head in orange). Blue and green lines represent sequences that connect in opposite directions (Tail to Tail in blue and Head to Head in green).

During the second mitotic divisions, the replicating micronucleus chromosome pulverizes and reassembles randomly, forming a shattered chromosome, which is then incorporated into the normal set [1].

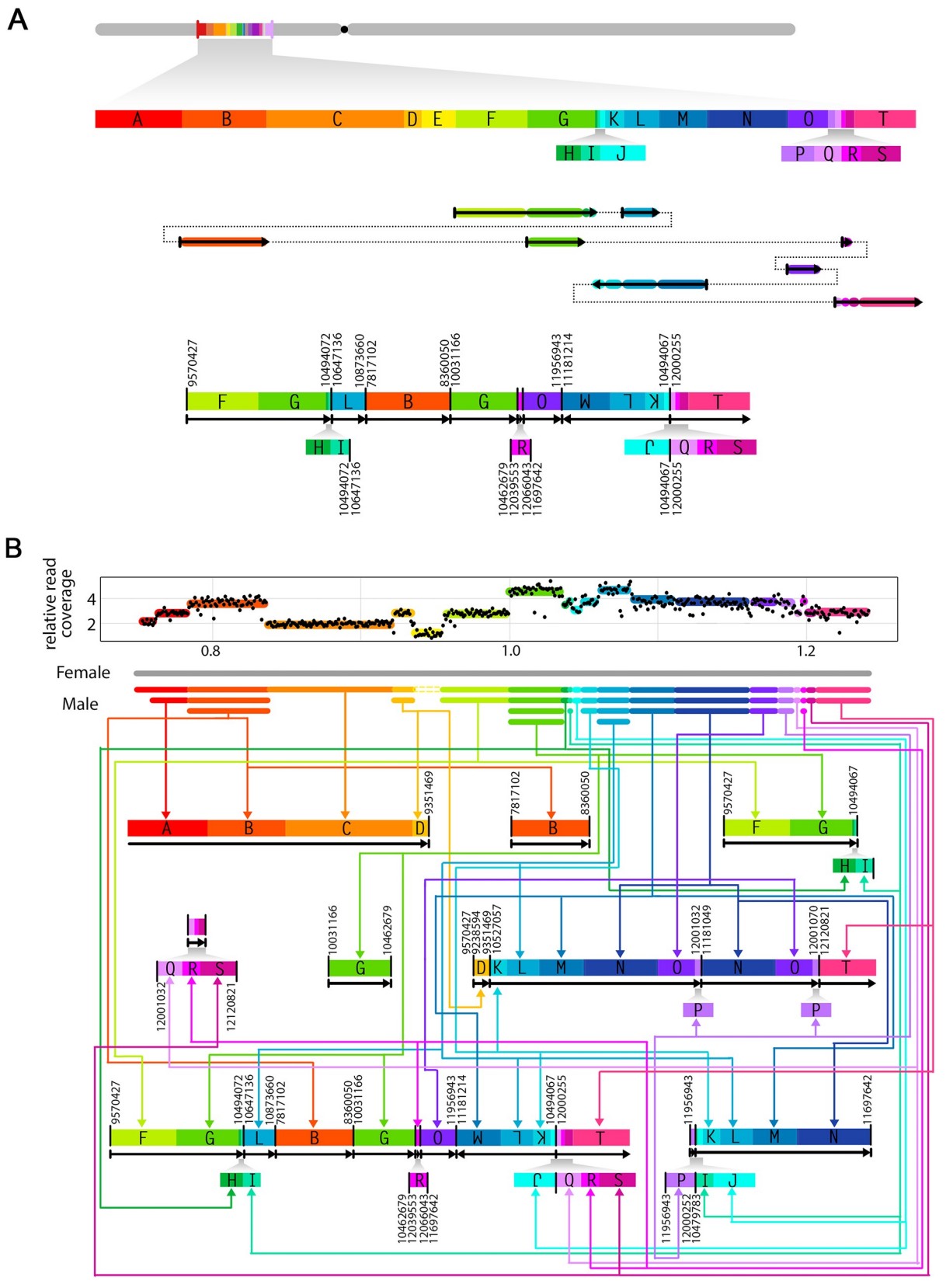

**Fig 6. Unraveling the structure of the shattered chromosomes.** Schematic diagrams illustrating the breakpoints rearrangement in one of the genome shattered lines (POP33_31). (A) The reference chromosome 1 is shown in grey and the regions engaged in rearrangement are labeled in alphabetical order. Each labeled block has a unique color and represents a genomic fragment with validated breakpoints on its flanking ends. The small blocks are enlarged below. All block sizes are proportional to genomic coordinates. In the middle is the potential junctions creating one of the rearranged fragments. Solid arrows with colors represent the corresponding blocks, and the dashed lines illustrate the order of blocks reconstruction. At the bottom is the new structure of that same fragment. Novel junctions are highlighted with bold vertical lines, and are labeled with their original genomic positions on two sides. Black arrows below blocks indicate the orientation of reconstructed fragments. Small blocks are enlarged below proportionally. Fragment duplications are linked and pointed out with the same color. (B) Summary of the fragments reconstructed based on the data obtained from line POP33_31. The dosage plot on top displays relative read coverage of the shattering region in chromosome 1. Each DNA block is labeled with the same color used in (A). A schematic representation of chromosome is shown below the dosage plot, with female (*P. deltoides*, WT) inherited chromosome colored in grey, and male (*P. nigra*, pollen irradiated) inherited chromosome illustrated in their corresponded colors and copy number states. For each DNA block, the same color arrows guide to the corresponding fragments on the reconstructed chromosome pieces.

In poplar, mature pollen is binucleate, and must undergo the second pollen mitosis, in which the generative cell divides into two sperm cells, just before fertilizing the egg cell. It is thus possible that the radiation-induced DNA breaks remained unrepaired, causing chromosome missegregation during the generative cell division, possibly resulting in the formation of a micronucleus in one of the sperm cells (Fig 8). After fertilization of the egg cell by the micronucleus-carrying sperm, during the first zygotic mitotic division, damage, such as incomplete replication, results in catastrophic DNA pulverization of the chromosome in the micronucleus [1,6]. The rearranged chromosome is reincorporated in the normal set during the subsequent mitosis. If the centromere is present, the shattered chromosome can segregate normally in the main nucleus, fixing the rearrangement.

Our study shows that novel DNA junctions were significantly enriched in gene-rich regions, which is consistent with Tan's results in Arabidopsis [14]. Similar outcomes have also been demonstrated in human breast cancer, where high density of DSBs occurred on chromosome 17, one of the human chromosomes with high gene content [32]. Further, open chromatin may be more available for recombination. In our analysis, 14/26 breakpoints formed junctions between genes, suggesting the potential of these events for genic innovation.

Our analysis used Illumina short reads to identify and assemble novel DNA junctions. In the shattered lines, this approach was successful as 78% of novel DNA junctions could be validated *in vitro*. Based on the number of copy number variation boundaries found in these two lines, and the number of novel breakpoints (each junction contains two breakpoints) that match these boundaries, we can estimate that we successfully identified 68% of the novel breakpoints. The false negative breakpoints could be caused by poor read coverage across those genomic regions, by the presence of repeated sequences complicating the read mapping process, or by differences between the reference genome used for read mapping (*P. trichocarpa)* and that of the male parent (*P. nigra)*. When applying this approach to sibling lines containing sparse indels along the genome, we did not identify any novel breakpoints despite the presence of seven large-scale insertions in these lines, which indicates that at least 7 new breakpoints should be present. As a result, it is still unclear where the duplicated fragments detected in these lines are located. Given that the probability of identifying real breakpoints in the two lines displaying shattering was 0.68 (34 breakpoints / 50 copy number shifts), our failure to find any real breakpoint out of 7 copy number shifts for the normal indels is surprising (p-value of Bootstrap hypothesis testing = 0.0081). It is thus possible that breaks giving rise to indels may result from a different DNA damage mechanism. For instance, unlike the junctions detected in the shattered lines, those present in the other lines may not be located within gene space and might therefore be more difficult to detect using short reads. Nevertheless, our analysis using short reads was successful at identifying enough novel junctions to confirm the randomly reorganized state of the shattered chromosomes.

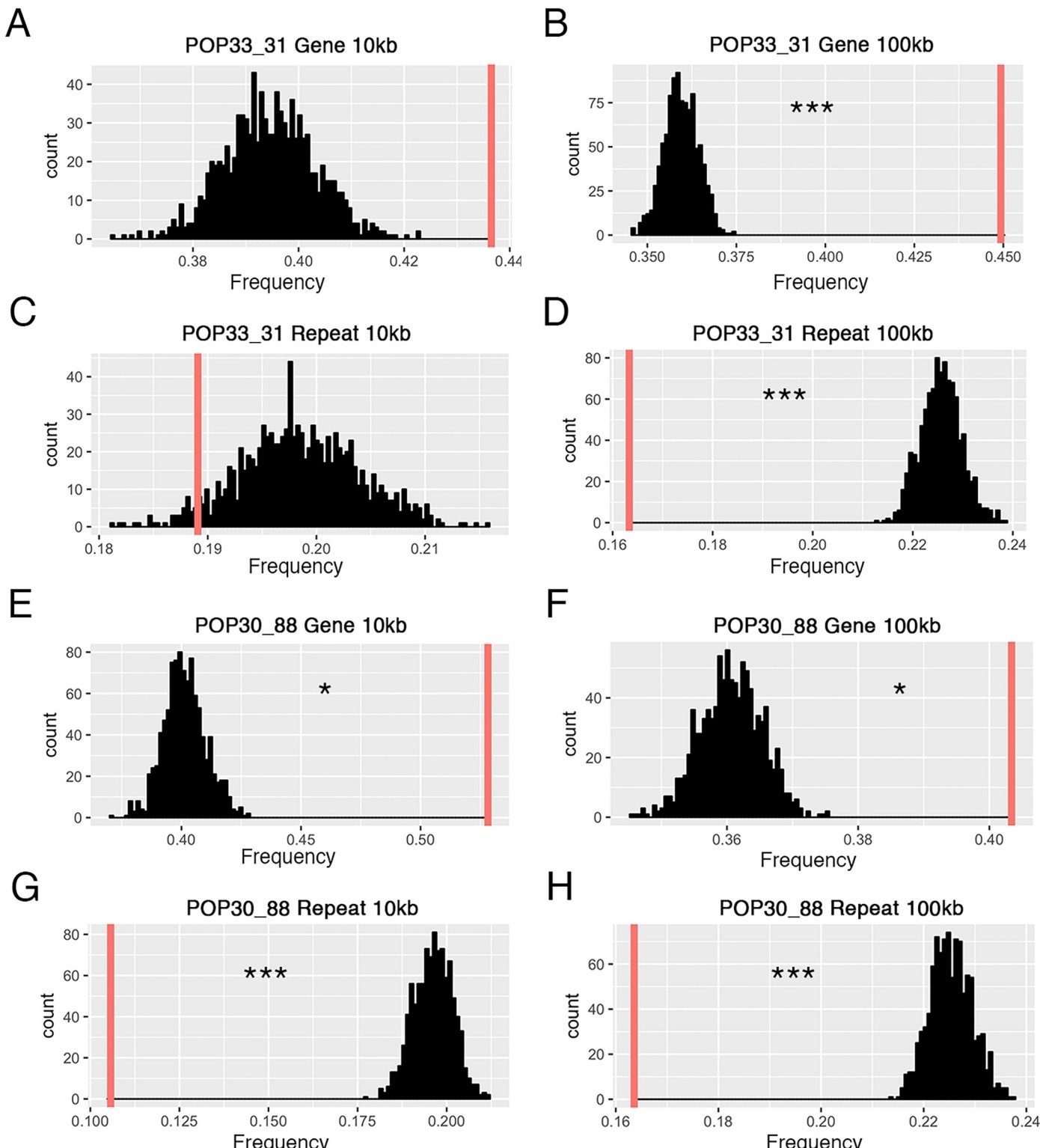

**Fig 7. Sequence context surrounding the breakpoints of novel DNA junctions.** The frequency of genes and repeated elements surrounding novel junctions is compared to the corresponding frequencies in randomly selected pseudo junctions. For each panel: 1,000 pseudo-junction were selected at random and the mean percentage of gene or TE space in these 1,000 junctions was calculated. This process was repeated 1,000 times and the distribution of these means are represented in black. The red vertical line represents the mean of enriched frequency for the observed validated novel junctions. Breakpoints of novel junctions in POP33_31 (A-D) occur significantly in gene-rich, repeats-deficient regions under 100kb window size (p-value < 0.001), but do not show statistical significance in 10kb

window size. Results were similar for POP30_88 (E-H). The observed junctions are significantly enriched with genes (p-value < 0.05), and have the lack of repeated elements (p-value < 0.001) regardless of window size.

In conclusion, our study demonstrated that chromoanagenesis can be induced in plants by ionizing radiation of pollen, indicating that extreme chromosomal rearrangements can be more widespread, and more tolerated than expected. Notably, natural mechanisms can also produce dsDNA breaks in pollen [16,17]. This type of cataclysmic outcomes is thus possible in a natural setting and can contribute evolutionary innovations, similarly to chromosomal inversions [33]. They may also mediate gene amplification [34], which has been detected in glyphosate-resistant weeds [35]. Because poplar is vegetatively propagated, we were able to produce several clones from each chromoanagenetic line and maintain some of these extreme chromosomal rearrangements in the field for at least five years. Finally, our results show that the observed chromosomal rearrangements directly affected the sequence of multiple genes and, in some cases, have the potential to produce new chimeric proteins. While most of these random events will probably result in non- or dys-functional proteins, it is an interesting avenue for the creation of new gene functions.

## Materials and methods

### Genomic sequencing and dosage variation analysis

Genomic DNA was extracted from leaf samples and prepared for deep-sequencing using Illumina technology, as previously described [19]. Sequencing reads (150 PE) were demultiplexed into individual libraries based on their barcodes, using a custom Python script (http://comailab.genomecenter.ucdavis.edu/index.php/Barcoded_data_preparation_tools), as described in previous studies [19]. Next, reads were aligned to the poplar reference *P. trichocarpa* v3.0 [36], using a custom Python script based on mapping using BWA [37] (http://comailab.genomecenter.ucdavis.edu/index.php/Bwa-doall). This generated an alignment file (sam file) for each line, which was used for further analysis.

To detect dosage variation, we calculated relative read coverage values across the genome for each line, as described previously [38]. Specifically, the genome was divided into a series of non-overlapping consecutive bins of 100kb or 10kb, depending on sequence coverage (see results). Next, for each bin, relative read coverage was calculated by taking the fraction of aligned reads in a particular bin for that line, and dividing it by the mean fraction of reads aligning to the same bin in all lines, and multiplying by 2, the background ploidy of poplar. A custom Python script was used to achieve these calculations (http://comailab.genomecenter.ucdavis.edu/index.php/Bin-by-sam). The relative coverage values obtained were then plotted according to the corresponded genomic region of their belonging bins. Values around 2 indicate the expected two copies, while values closer to 3 and 1 suggest the presence of insertions, or deletions, respectively.

### Detection of novel genomic junctions

To detect novel genomic junctions, we first searched for indels boundaries, which represented the potential breakpoints of reorganized genomic fragments. Based on the dosage variation plots obtained using 10kb bins, we recorded potential junctions using the following criteria: bins where relative read coverage decreased or increased by >0.7 compared to their adjacent forward bin, and instances where this trend was true for at least three consecutive bins. Additionally, potential breakpoints were only retained if they were unique to a single line. These potential breakpoints became the most likely locations for forming novel DNA junctions. To

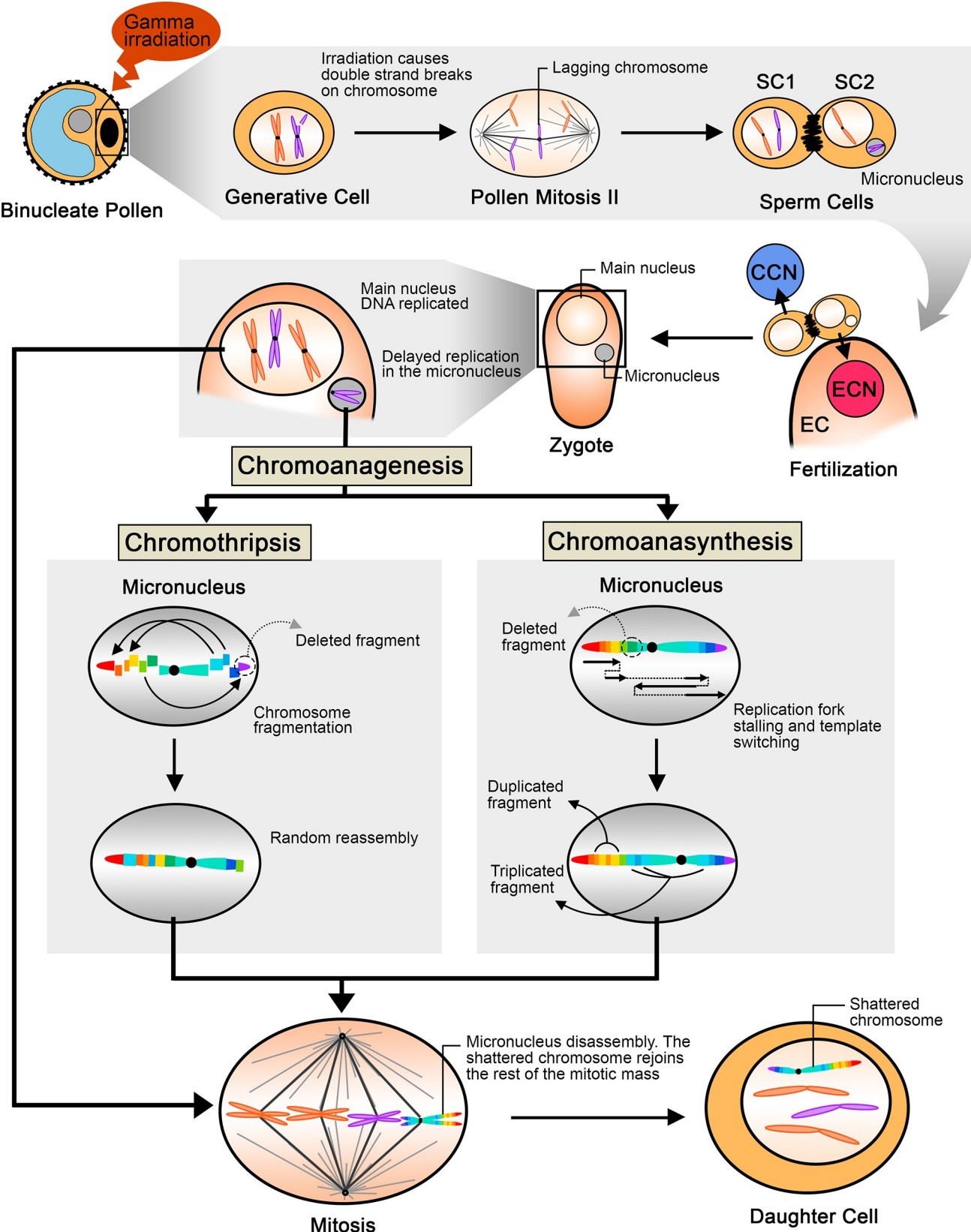

**Fig 8. Proposed model illustrating the steps leading to chromoanagenesis following pollen irradiation.** Gamma irradiation of binucleate pollen induces double stranded DNA breaks in the generative cell, and results in chromosome lagging or in bridge formation [12] during the second pollen mitosis. The lagging chromosome is excluded from the main nucleus and forms a micronucleus. The sperm cell carrying the micronucleus undergoes karyogamy with the egg cell, and produces a zygote with a (2n-1) nucleus and a micronucleus containing a single paternal chromosome. DNA replication in micronuclei is delayed and leads to chromoanagenesis via two possible mechanisms, chromothripsis and chromoanasynthesis, which were both observed in our poplar lines. Chromothripsis involves fragmentation and random reassembly, while chromoanasynthesis results from replication fork stalling and template switching. The highly rearranged chromosome is eventually released from the micronucleus and reunites with the main nuclear genome during mitotic division. The shattered chromosome is thereafter retained in the main nucleus. SC: sperm cell; EC: egg cell; ECN: egg cell nucleus; CCN: central cell nucleus.

characterize novel junctions in more detail, we next searched for reads mapping to two distant genomic locations, and therefore crossed the targeted junctions. A custom Python script (https://github.com/guoweier/Poplar_Chromoanagenesis) was used. Specifically, the script divided the genome into non-overlapping consecutive 10kb bins and, for each combination of two non-consecutive bins that were at least 2,000 bp apart, the number of reads mapping to both bins was recorded for each line. Numbers were then compared between lines to identify pairs of bins with high coverage in a single line compared to the others, suggesting the presence of a novel junction. In order to set a minimum threshold of coverage to eliminate false positives, we needed to calculate the expected average coverage over each junction. To do so, we created artificial non-overlapping 5 kb bins throughout the genome, considered the boundary between two consecutive bins as pseudo-junctions and recorded the average reads coverage at these pseudo-junctions. These values were then divided by 2, to account for the fact that these pseudo-junctions are expected to be present in two copies in the diploid poplar genome, while the indels and other novel junctions are expected to only affect one copy of the genome. We used these line-specific thresholds as minimum coverage thresholds for the identification of potential novel junctions. Second, to ensure that junctions were specific to a single line, we discarded bin-pairs that were positive in more than one line. Specifically, a potential in-pair was only retained if none of the other lines exhibited reads that mapped to those two bins.

## Novel junction validation

To assemble potential novel junctions, we searched the alignment file (sam file) of each line and extracted the cross-junction reads identified at the selected bins, using a custom Python script (https://github.com/guoweier/Poplar_Chromoanagenesis). Next, the PRICE genome assembler was used to assemble the cross-junction reads into contigs [39]. The assembly parameters and input data can be found in our github repository (https://github.com/guoweier/Poplar_Chromoanagenesis). To confirm the junction genomic composition, we aligned the output contigs to the *P. trichocarpa* genome by using blast+ package [40] by using a custom bash script (https://github.com/guoweier/Poplar_Chromoanagenesis). When the two ends of the contig aligned to the expected regions, we considered that the novel junction was confirmed *in silico*.

To validate these potential junctions *in vitro*, PCR primers were designed using Primer3 [37] (S1 Table). PCR were run using the GoTaq Green Mastermix (Promega Corporation, Madison, WI) with 1ng sample gDNA. The obtained PCR products were purified using gel extraction (QIAquick Gel Extraction Kit, Qiagen) and sent for Sanger sequencing.

## SNP frequency analysis

We used parental SNP allelic percentage to identify the parental origin of the lesions. Single nucleotide polymorphism (SNP) between *P. deltoides* (female) and *P. nigra* (male) were identified previously [19]. We genotyped each line as described previously [19]. In short, to calculate

the percentage of *P. nigra* and *P. deltoides* alleles at each position, we created an mpileup file containing every base allele and coverage for all examined lines, using a custom Python package based on Samtools [41] and built-in mpileup function (http://comailab.genomecenter. ucdavis.edu/index.php/Mpileup). The mpileup file was then simplified by converting a parsed-mpileup file, using the custom Python package described above. Next, the parsed-mpileup file was used to search for the preselected SNPs position. Finally, to obtain robust allele percentages, SNP allele calls were pooled within consecutive bins, and the percentage of *P. nigra* parental alleles were calculated for each bin. According to this approach, a diploid chromosome exhibited 50% *P. nigra* alleles. A deletion on one chromosome is expected to exhibit 0% or 100% *P. nigra* alleles, depending on which parental chromosome was lost. An increase of copy number states is expected to exhibit allelic ratio bias between two parents, with 1:2 represented DNA fragment duplication, 1:3 represented DNA fragment triplication, and so on.

### Genome restructuring analysis

To reconstruct each mutant genome based on the identified validated junctions, we searched for fragments with the same breakpoints and strung them together manually, with the expectation that each breakpoint should be involved in two junctions, one on each side of the breakpoint. With this logic, we manually looked for paired fragment end locations among the junctions, and arranged them into longer pieces. We then built the rearranged chromosomes, while taking junction orientation and fragments copy number into account.

### Enrichment ratio analysis

The poplar genome annotation file, including the genomic positions of gene and repeatmasked (GFF-Version3.0) was downloaded from Phytozome (http://phytozome.jgi.doe.gov/pz/portal. html). Next, we used a custom python script to calculate genes/repeats density around each of the novel breakpoints (https://github.com/guoweier/Poplar_Chromoanagenesis). Specifically, each potential breakpoint was set as the center of a 10kb or 100kb window, and the nucleotide number of typical genomic features within these windows was recorded. Next, to provide a random set of junctions, we used the previously constructed pseudo-junction pool, and randomly selected 1,000 of these pseudo breaks for genomic feature density calculation. For each line, this type of random pseudo-break datasets were established 1,000 times for every examined genomic feature. Enrichment ratios were calculated by taking the means of genomic feature density at real breakpoints, divided by the means of the corresponded features density at random pseudo breakpoints datasets. Significance was assessed by comparing the density of real breakpoints and 1,000 randomized datasets using one sample t-test.

## Supporting information

**S1 Fig. Summary of the fragments reconstructed on line POP30_88 Chromosome 2.** The diagram follows the same criteria as in Fig 6B. The rearranged fragments were constructed based on the data of novel junction observation in POP30_88.
(TIF)

**S2 Fig. Detailed view of the *P. nigra* SNP frequency pattern in the shattered regions.** The genome was divided into consecutive non-overlapping 10kb bins. Each blue dot represents the average *P. nigra* SNP frequency for a 10kb bin. Horizontal lines exhibit the expected frequency levels for different copy number states, with their numbers labeled on the right.
(TIF)

**S1 Table. PCR primers used in novel junction validation.** List of primers used for PCR amplification.
(XLSX)

**S2 Table. Summary of indels in 2 shattered lines.** Large-scale indels in two shattering lines (POP33_31 = 21, POP30_88 = 11) were identified based on dosage variation patterns and SNP frequency. The locations and copy number states of indels are indicated, as well as the parental genotype they originated from. D: *P. deltoides*; N: *P. nigra*.
(XLSX)

**S3 Table. Summary of all validated novel DNA junctions.** List of validated novel DNA junctions in the two shattering lines (POP33_31 = 12, POP30_88 = 14). Each junction is indicated with its junction type, two breakpoints positions, orientation, and its correlation with CNV edges.
(XLSX)

**S4 Table. DNA context at the breakpoints.** List of all breakpoints identified, as well as information about the affected genes when the breakpoints occurred within a gene.
(XLSX)

**S5 Table. Summary of the possible gene fusion events at novel DNA junctions.** List of junctions containing gene to gene fusion within the two shattering lines. Instances where two genes are fused in the same direction are labelled in green, indicating that these fusions might form novel gene products.
(XLSX)

**S6 Table. DNA context surrounding the novel junctions.** The novel DNA junctions identified in the two lines (POP33_31 = 12, POP30_88 = 14) exhibiting clustered patterns are preferentially located in regions that are rich in gene sequences and poor in repeated sequences, compared to the rest of the genome. Enrichment ratio represents the comparison between the means of genomic feature density at real breakpoints and the means of a similar set of randomly selected features. Ratio >1 indicates validated breakpoints that have a higher density of features than the genome average, while ratio <1 indicates the lower density of validated breakpoints compared to the genome. Genes are enriched near breakpoints of both lines with 100kb-window size (POP33_31 $p<0.001$; POP30_88 $p<0.05$). The lack of repeated elements surrounding breakpoints are observed in both lines as well (POP33_31 $p<0.001$; POP30_88 $p<0.001$).
(XLSX)

## Acknowledgments

We acknowledge Meric Lieberman for assistance on bioinformatics.

## Author Contributions

**Conceptualization:** Weier Guo, Luca Comai, Isabelle M. Henry.

**Formal analysis:** Weier Guo, Isabelle M. Henry.

**Funding acquisition:** Luca Comai, Isabelle M. Henry.

**Investigation:** Weier Guo.

**Methodology:** Weier Guo, Isabelle M. Henry.

Project administration: Isabelle M. Henry.

Software: Weier Guo.

Visualization: Weier Guo.

Writing – original draft: Weier Guo.

Writing – review & editing: Weier Guo, Luca Comai, Isabelle M. Henry.

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
