## [Decision Letter · Decision Letter 0]

8 Jun 2021

Dear Dr Henry,

Thank you very much for submitting your Research Article entitled 'Chromoanagenesis from radiation-induced genome damage in Populus' to PLOS Genetics.

The manuscript was fully evaluated at the editorial level and by independent peer reviewers. The reviewers appreciated the attention to an important topic but identified some concerns that we ask you address in a revised manuscript

We therefore ask you to modify the manuscript according to the review recommendations. Your revisions should address the specific points made by each reviewer.

[LINK]

Yours sincerely,

Ian Henderson

Associate Editor

PLOS Genetics

Claudia Köhler

Section Editor: Plant Genetics

PLOS Genetics

Reviewer's Responses to Questions

**Comments to the Authors:**

Reviewer #1: The use of pollen irradiation has been investigated for many years, mainly for the purpose of haploidization and the generation of mutants. While traditional chromosome rearrangements have been described, the range of variation triggered by DNA breakage and repair, has not been determined. The authors performed detailed sequence analysis of several M1 poplar plants and identified individuals with genomic patterns reminiscent of chromoanagenesis. The data suggest that fragmentation and repair of DNA can produce extreme chromosome variations.

Considering that the repair of DNA breaks likely also occurs after zygote formation, I’m wondering whether the resulting M1 plants are chimeric. A chimeric situation could influence the outcome of the sequence analysis. To address this possibility, the authors should analyze genomic DNA isolated from individual branches of the same tree. A simple PCR with primer combinations suitable to identify rearranged DNA fragments could be used to address this concern.

Reviewer #2: This is a very nice and timely contribution to the emerging topic of chromothripsis, more correctly chromosome anagenesis, in plants. I have only minor suggestions and one or two major questions. I congratulate to the authors to an elegantly conducted and written report.

Author summary

„They“. Not clear if this relates to „mechanismS“ or to „type of rearrangement“.

Helps TO understand.

Genome instability….genome evolution (1x genome enough?)

If alone standing, it is unclear what „provides a new system to characterize these types of changes.” ´means. Actually, the last sentence would benefit from some more careful revision.

Please unify throughout: chromosome or chromosomal rearrangement.

Introduction

Although the facts are well set, I am asking the authors to revise the wording in places.

- I was not really sure what to imagine under „clustered copy number variation“. As this term is used further on, I suggest to define a little what is understood under „clustered“ (and/or „unclustered“) CNV. This can also be done by referring to an example.

- „The extreme restructuring of a single chromosome (or rarely two or more)” this is later contradicted by the definition of chromoplexy (I mean single vs. two or more)

- Chromoanagenesis originates from a single event (as it is written), the description of chromothripsis follows this logic by (one) „dsDNA breaks“. However, one break will not result in rearrangements of tens to hundreds segments. I consider the definition of chromoanagenesis as a single event to be most critical. While I understand the point, chromothripsis usually results from a number of events (i.e. DSBs) caused (probably) by one (major) agent/stimulus.

I was unsure whether I understand what SHUFFLED chains of rearrangements mean.

I do not understand to what „together covering the whole genome multiple times [19].” is related. To insertions and deletions?

Results

Please explain better how you came from 2 trees to 9 plants. This is somewhat confusing at present.

“Genomic DNAs were sent”

“individuals originated from loss or gain of the paternal P. nigra copy (Fig 2), confirming that pollen irradiation caused dosage variation” Although I understood the point, some readers may not be so clear that P. nigra donated the paternal genome in previous crosses.

“more than 10 CNVs clustered” Then, 21 CNVs…Does this simply mean that 21 is more than 10? If yes, perhaps a range would be better…

When starting to describe where CNVs were located, it would be good to say how many chromosomes the poplar hybrids had. This of course could be stated earlier in the text. Also, a reader is informed that the hybrids are sterile (is this really true or the trees were too young to flower?), but there is no information as why? Is this due to problems with chromosome pairing (mitosis, meiosis), whas this researched?

Unify throughout: shattering group OR Shattering Group.

At the end of this paragraph…it is a bit diffucult to follow the narrative. The difference between the two plants are insertions vs. duplications. Please make clearer why “different” the mechanisms were different. Also, I suggest to be more specific in this section. For example, I was not able to easily see how big the insertions were compared to deletions. Another point is figure 2. If looking at fig. 2A (for instance), how can I see 2 deletions and 19 insertions? And copy number states? At least some of the events could be colored or emphasized in some other way.

Extreme genomic rearrangements are associated with intra-chromosomal junctions

Referring to fig. 4D does not comply with the definition of chromoanagenesis (two different chromosomes).

“Additionally, 26 (50%) breakpoints affected a gene coding sequence directly” I wonder whether the authors cannot be a little more specific what “affected” means. Considering that deletions, insertions and inversions were documented, it would be nice to spend more time on specifying the effects of these rearrangements on gene sructure/function.

Discussion

“For those novel DNA junctions, 1-11bp overlapping sequences were found between the joined fragments,” I understood that this was NOT always the case (see Results).

Discussion on pages 14+15 is not very clear (but important):

I read that a sperm cell cotains a micronucleus, then it is said that “during the first zygotic mitotic division, damage, such as incomplete replication, results in catastrophic DNA pulverization of the chromosome in the micronucleus”. What does this mean? Is the meaning that the micronucleus is a transient state in the sperm cell and that the same chromosome forms a micronucleus repeatedly? This is explained in the next two sentences, but not exactly. Also, here and earlier in the text, I was not clear on how chromoanagenesis of a single chromosome in a single (parental) genome influences pairing of homologous chromosomes? Is there data for it?

Please add your thoughts on why only the two chromosomes were affected, not other. Do these two chromosomes differ from the remaining ones? Are the chromatin profiles of the two chromosomes somewhat different from the remaining ones? Such an analysis should be carried out.

Perhaps I overlooked this in Discussion. However, I wonder why DSBs occurred in gene-rich regions in poplar and arabidopsis. Some tentative explanation in Discussion would further improve the paper. Does this mean that, for example, repetitive arrays are more resistant towards demaging gamma radiation? I am almost certain that some literature is available on the topic.

Figure 8.

Why chromothripsis, but mainly chromoanagenesis, is not thought to occur already during DNA replication before pollen mitosis II?

Martin Lysak

Reviewer #3: Guo et al provide a thorough description of two cases of chromoanagenesis-like events in poplar, apparently induced by gamma irradiation. One looked more like chromoanasynthesis, and another more like chromothripsis, but both cases involved single chromosomes, suggesting a mechanism involving micronuclei formation, partial degradation of the micronuclei, and ultimate reincorporation into the main nucleus. This is of interest because the events occurred in pollen and in lines that had been irradiated, making the work relevant to future mutagenesis studies.

The paper is relatively simple and follows a path well developed by Henry and Comai using short read alignment, some local assembly and PCR validation. Likewise the figures show dosage variation in a familiar way and review the chromothripsis (and variations) literature appropriately. The plant literature is small and well covered here. The paper is well written. Figure 8, showing potential mechanisms, is very nicely done. I enjoyed reading the paper.

**Have all data underlying the figures and results presented in the manuscript been provided?**

Reviewer #1: Yes

Reviewer #2: Yes

Reviewer #3: Yes

PLOS authors have the option to publish the peer review history of their article (what does this mean?). If published, this will include your full peer review and any attached files.

Reviewer #1: No

Reviewer #2: **Yes: **Martin A. Lysak

Reviewer #3: No

---

## [Decision Letter · Decision Letter 1]

22 Jul 2021

Dear Dr Henry,

We are pleased to inform you that your manuscript entitled "Chromoanagenesis from radiation-induced genome damage in Populus" has been editorially accepted for publication in PLOS Genetics. Congratulations!

Yours sincerely,

Ian Henderson

Associate Editor

PLOS Genetics

Claudia Köhler

Section Editor: Plant Genetics

PLOS Genetics

Comments from the reviewers (if applicable):

Reviewer's Responses to Questions

**Comments to the Authors:**

Reviewer #1: I'm satisfied with the revised version of the manuscript.

Reviewer #2: I am happy with the changes reflecting on my comments/suggestions.

Reviewer #3: I looked over the responses to the other reviewers and they all seem thoughtful and appropriate. It was a good paper before and now it is further improved.

**Have all data underlying the figures and results presented in the manuscript been provided?**

Reviewer #1: Yes

Reviewer #2: Yes

Reviewer #3: Yes

PLOS authors have the option to publish the peer review history of their article (what does this mean?). If published, this will include your full peer review and any attached files.

Reviewer #1: No

Reviewer #2: No

Reviewer #3: No

**Data Deposition**

http://datadryad.org/submit?journalID=pgenetics&manu=PGENETICS-D-21-00625R1

**Press Queries**

---

## [Editor Report · Acceptance letter]

20 Aug 2021

PGENETICS-D-21-00625R1 

Chromoanagenesis from radiation-induced genome damage in Populus 

Dear Dr Henry, 

We are pleased to inform you that your manuscript entitled "Chromoanagenesis from radiation-induced genome damage in Populus" has been formally accepted for publication in PLOS Genetics! Your manuscript is now with our production department and you will be notified of the publication date in due course.

With kind regards,

Zsofi Zombor

PLOS Genetics

On behalf of:
